# Development of an ICT-Based Exergame Program for Children with Developmental Disabilities

**DOI:** 10.3390/jcm11195890

**Published:** 2022-10-05

**Authors:** Hyunjin Kwon, Hyokju Maeng, Jinwook Chung

**Affiliations:** 1Department of Sport Culture, Dongguk University, Seoul 04620, Korea; 2Department of Kinesiology and Health, Georgia State University, Atlanta, GA 30093, USA

**Keywords:** ICT, exergame, developmental disability, intellectual disability, autism spectrum disorder

## Abstract

The purpose of this study was to develop an information and communications technology (ICT)-based exergame for children with developmental disabilities (DD) and to examine its impacts on physical fitness and fundamental motor skills (FMS). The ICT-based exergame consisted of visual and auditory demonstrations of diverse locomotor movement and object manipulation activities by the virtual characters. A total of 52 children with DD participated in the present study. The participants were divided into twenty-seven children in the experimental group and 25 children with DD in the control group. The experimental group participants engaged in the exergame program for 12 weeks. All participants’ muscle strength (i.e., standing long jump) and four fundamental motor skills, such as the horizontal jump, hop, overhand throw, and dribble, were assessed during the pre- and post-test process. There were significant impacts on physical fitness and FMS (*p* < 0.001) between the groups. Specifically, the results of three FMS (hop, overhand throw, and dribble) and standing long jump significantly improved in children with DD except for the horizontal jump skill. The results of this study is evidence that the ICT-based exergame program for children with DD may be utilized to improve physical fitness and FMS in children with DD.

## 1. Introduction

Exergame is a compound word for exercise and game. This term is defined as technology-driven physical activities, such as video game play, that require participants to be physically active or exercise to play the game. These games require the user to apply full body motion to participate in virtual sports, group fitness exercises, or other interactive physical activities. The concept of exergame takes the passion for gaming and turns what was once considered a sedentary behavior into a potentially more active and healthy activity [1]. Virtual reality (VR) and augmented reality (AR) have arisen in the field of exergames. VR creates an immersive virtual environment, while AR augments a real-world scene. VR requires a headset device, while AR does not. The Nintendo Wii^TM^, Microsoft Kinect^TM^ (Redmond, WA, USA), and mobile phone applications, such as Pokémon Go^TM^, were developed to play exercise and physical activity in multiple forms of exergame. Exergames can be utilized as a potential rehabilitation tool for increasing physical activity and improving health status. The technology of VR (i.e., exergame) has been initiated to utilize enhancing functional abilities of persons with disabilities due to its impact on functional benefits with joyful engagement in tasks or activities for the past decade [2,3]. Especially, the benefits of exergames were studied to investigate their impacts on physical activity [4,5], motor function [6,7,8], and health [9,10].

Developmental disability (DD) is an assorted group of conditions that affects the trajectory of the individual’s physical, intellectual, or emotional development, influencing their personal, social, academic, or occupational function [11,12]. There are multiple types of developmental disabilities, such as intellectual disability (ID), autism spectrum disorder (ASD), Down syndrome, and cerebral palsy. Children with DD show unique characteristics due to a limitation in physical, learning, language, or behavioral areas [9]. Children with DD may have limitations in social interaction, verbal and nonverbal communication, and creative activity with interests, cognition, fundamental motor skills (FMS; i.e., gross motor skills), physical fitness, and behavior patterns [12]. Long-term issues may include difficulties in creating and maintaining relationships, retaining a job, and performing daily tasks [13].

Moreover, diverse studies have been implemented to enhance the performance of motor skills required to promote physical activity. In particular, poor performance levels in locomotor and object manipulation skills for actively engaging in physical activity were reported in studies of the development of gross motor skills among children with DD. Children with DD tend to be significantly delayed in their FMS compared with typically developing peers [14,15,16]. Several studies have shown that children with ID [17,18] and ASD [16,19,20,21] tended to demonstrate delayed FMS development. According to Westendorp and colleagues [17], motor delays among children with ID appeared in motor performance requiring greater speed and movement control. By all accounts, the characteristic of children with ASD in motor development was similar to students with ID. Common, is a delay of FMS among those populations [16,22,23].

The prevalence of ASD is reported to be 1 in 54 people. and research is being focused on improving their physical activity [24]. This was because the physical activity level of children with ASD does not follow the latest physical activity guidelines [25,26]. Case and colleagues [25] found that only 14.2% of children with autism between the ages of 6 and 17 met the 60-min daily physical activity guidelines. In addition, McCoy, Jakicic, and Gibbs [27] reported that children with ASD were 60% less likely to participate in PA. This study additionally presented that they were 74% less likely to participate in PA compared to typically developing children. This inactivity may be one of the largest contributors to the increased prevalence of excess weight or obesity. Perhaps of greater concern is that children with DD including ID and ASD have substantially higher levels of obesity [28] and lower levels of physical activity (PA) [29,30] compared to their peers without disabilities.

More specifically, children with ID between the ages of 3 and 10 years engaged in less PA than their typically developing peers [31]. Children with ASD are also less likely to reach the recommended MVPA levels compared to their typically developing peers [32]. With increasing age and the transition to adolescence, their participation in PA was further reduced in amount and intensity [33,34,35]. Participation in low levels and intensities of PA can exacerbate problems of obesity and poor physical health, highlighting the importance of promoting lifelong participation [36]. For children of all abilities, participation in sustained bouts of PA contributes to the promotion and maintenance of health and physical fitness [37]. Agiovlasitis and colleagues [38], argued that a complex effort is required to consider the interaction characteristics including individual and environmental factors to promote physical activity. However, only solid characteristics among individuals with DD, such as a lack of individual motor performance level [39] or lack of social skills [40,41] have been considered in those studies. Those behavioral and movement challenges might relevantly describe the characteristics among children with ID and ASD among various types of disabilities in DD.

Recently, exergame has been actively disseminated for physical activity improvement and exercise performance skills development by inducing the disabled to participate in physical activities. Based on the potential goal of health promotion, exergames were utilized to improve physical fitness as well as develop fundamental motor skill performance of the body. However, previous studies pointed out that existing exergames had limitations to engaging and performing properly due to the difficulty of playing for persons with disabilities, especially for moderate and severe cognitive disabilities [7,8]. The design of exergames for use by individuals with disabilities requires consideration of barriers to access such as operating the game interface, perceiving game events, and production of movements required for gameplay as well as specific benefits and risks. It is essential to involve players such as individuals with disabilities, family members, and clinicians who may interact with the person and the games [42]. Specifically, Wiemeyer and colleagues [43] presented developing optima exergame designs and targeted interventions for individuals with disabilities that require a multifaceted approach. Accordingly, this researcher judged that developing an easy, fun, and practical exergame program is necessary considering the cognitive and psychodynamic characteristics of children with DD. The purpose of this study was to develop an exergame program for children with DD and to verify its effectiveness. The exergame program application results developed in this study could be used as meaningful data for future exergame program development for children with DD.

## 2. Materials and Methods

### 2.1. Developing an ICT-Based Exergame for Children with Developmental Disabilities

The development process could be broadly divided into planning, development, and effectiveness verification stages. Reference study reviews related to the behavior and movement characteristics of children with DD as well as exergame implementation for those populations were conducted to arrange findings, considerations, and suggestions for the information and communications technology (ICT)-based exergame program. Ten experts were involved in the research team for developing a new ICT-based exergame program for children with DD. They specialized following areas: (a) adapted physical education (*n* = 2), (b) motor learning and control (*n* = 2), (c) exercise physiology (*n* = 2), (d) psychology (*n* = 1), (e) occupational therapy (*n* = 1), and (f) computer science (*n* = 2).

Many studies presented that physical activity or exercise programs act as important factors that could directly influence the content level and effectiveness of it [44]. According to Wiemeyer and colleagues [43], adaptations and considerations should be applied to develop the ICT-based exergame program for children with disabilities according to their cognitive, affective, and psychomotor characteristics. Those components were important to designing the ICT-based exergame program. Specifically, the exergame mechanics should be secure to understand the required task activities for children with DD considering their developmental characteristics in cognition and physical performance. Also, the exergame interfaces were embedded so that children with DD could conduct task performance with their concentration, such as controls of arms, legs, and whole body in modified levels of task. Task modifications could be used flexibly in the size of targets, distance from the target, and speed of required performance according to the individual development of performance. The ICT-based exergame for children with DD should be joyful and fun with visual and auditorial resources to complete the given tasks [45,46,47].

The ICT-based exergame in the present study focused on providing appropriate task activities to improve fundamental motor skills and physical fitness among children with DD. A preliminary evaluation process was executed to decide what contents in the exergame program would be necessary and proper in the three domains, such as psychomotor, cognitive, and attractive, for children with DD. The research team established the evaluation subdomain on each domain considering the developmental and performance characteristics of children with DD as well as their interests (see Table 1). Also, adapted tasks were applied to enhance certain subdomains. For instance, finding an object (e.g., flower, butterfly) was used to induce visual perception processing in the cognitive domain.

Behavioral characteristics of children with DD were deliberated to make evaluation methods in the attractive domain. For example, catching a water drop was selected to investigate the development levels of those populations in the attractive domain. Regarding the psychomotor domain, the run, horizontal jump, hop, overhand throw, and dribble skills were examined to evaluate the development levels of coordination and motor skill among children with DD. The primary research established the fusion index with eight types of task activities according to the development levels in cognitive, attractive, and psychomotor areas in those populations. The difficulty levels of performance tasks were applied to different distances, sizes, speeds, and times for each participant. The performance tasks consisted of jumping, running, rhythm, overhand throwing, kicking, gymnastic, and imitation activities. Previous studies determined that the level of motor skill performance and physical fitness had a significant relationship among children with [48,49] and without disabilities [50,51]. This implied physical activity interventions could influence both motor skill competency and physical fitness in children with and without disabilities. Individualized ICT-based exergame programs fitted with the evaluation results of each child with DD were implemented to enhance their physical fitness and fundamental motor skills throughout the intervention period.

#### 2.1.1. Physical Fitness

For children with DD, physical fitness activities should be designed to consider their physical characteristics that were overweight, lack of muscle strength, endurance, and cardiovascular. Based on the selected motor skills, the exergame program required repeated skill performance according to the virtual character’s instructions. For example, three times jumping over the moving bar or chasing a visual subject on the ground. The ICT-based exergame program used various colors, sizes, and speeds of virtual resources to sustain the performance of children with DD in the activity setting (see Figure 1).

#### 2.1.2. Fundamental Motor Skills

Locomotor and object manipulation skills play a significant role in engaging in physical activities, such as jogging, sports, and game activities. The exergame program should be established with fun and joyful activities to perform gross motor skills using visual and auditory technologies. A present study applied the jump skill among various locomotor skills considering the development and performance level of children with DD. Jumping activities were created based on the ideas of jump rope, high jump, and children’s gameplay. Figure 1 showed a simulation and a real implementation of the jump activity. Regarding object manipulation skills, the researcher selected throw and kick skills which were widely used in sports and game activities. Also, those skills were appropriate to realize exergame activities in the minimized virtual technology setting compared to other skills. Performing throw and kick skills had targets, such as throwing a bean bag twice toward the target (i.e., flower, butterfly), or kicking a ball to desired soccer goals. The level of difficulty was controlled by the size and number of targets (see Figure 2). All desired activity performances were explained in the demonstration by the virtual character. Overall, easy and simple movement skill tasks were designed to play the exergame program for children with DD.

### 2.2. Setting

An exergame activity was implemented in a certain classroom size space that was 5 m (width), 5 m (length), and 4 m (height). Virtual reality projections, speakers, personal computers, and monitors were used to generate visual and auditory simulations in the space. All walls and the ground surfaces in the room were colored white to show screen effects for simulations in the ICT-based exergame program. All backgrounds of task activities were covered with natural landscapes containing mountains, sky, trees, and rocks to provide a relaxed feeling for children with DD.

### 2.3. Examining the Impact of ICT-Based Exergame Program

#### 2.3.1. Participants

Participants were children with developmental disabilities in South Korea. The inclusion criteria were as follows: (a) diagnosis of developmental disabilities, (b) age 7 to 12, and (c) no limitation to perform physical movement. A priori sample size was calculated using G*Power (version 3.1.9.7; Franz Faul, Kiel, Germany) to support the detected significance in the results. The required sample size was 36 participants with a power of 95% and an alpha value of 0.05 for the present study. The author recruited participants through website advertisements such as public and special schools, and local welfare centers. After explaining the experimental purpose, procedures, risks, and benefits of this study to the participants and their parents, the author obtained written informed consent from the parents of the participants. A total of fifty-two children with DD participated in the present study (see Figure 2). Twenty-seven children with DD (female *n* = 8, mean age ± SD = 10.15 ± 1.51) engaged in the ICT-based exergame program as the experimental group. For the control group, 25 children with DD (7 females, mean age ± SD = 10.24 ± 1.48) participated in the present study. Twenty-seven participants in the experimental group consisted of 15 children with autism spectrum disorders (ASD) and 12 children with intellectual disabilities (ID). Regarding the control group, children with DD were 15 ASD and 10 ID, respectively. Table 2 described the demographic characteristics of the participants in this study including height, weight, and calculated body mass index (BMI).

#### 2.3.2. Muscular Strength Test

Standing long jump is one of the measurement methods to evaluate explosive strength and is being used in various fields for physical fitness evaluation. Some studies used a standing long jump test to assess muscular strength in physical fitness among individuals with disabilities. Rintala et al. (2016) [52] measured the standing long jump among muscular strength tests in individuals with intellectual disabilities to evaluate their level of physical fitness. According to Yilmaz and colleagues (2017) [53], a standing long jump test was used to examine the effect of a 20-week exercise program on physical fitness in children with disabilities. Both studies used a traditional tape measurement method on the ground. Whereas the present study measured participants’ standing long jump using the markerless motion capture system ‘Kinect’ and active remote sensing systems ‘Intel Lidar’. A previous study measured a long jump in place using a Kinect sensor for children and compared it with the traditional method, which showed a 94% significant value. The Kinect assessment system consisted of Azure Kinect and Intel Lidar to measure accurate data of movement performance (see Figure 3). A primary researcher showed the participants a standing long jump video and demonstration of the performance to help them understand. Then, two or three practice trials were given to each participant considering their understanding condition for the standing long jump performance. The actual performance data were collected to use the highest record among the three trials of performance.

#### 2.3.3. Gross Motor Development Test

The Test of Gross Motor Development- Third edition (TGMD-3) is widely used to measure the fundamental motor skill development (FMS) of children aged 3 to 10. A total of 13 FMS skills falls under six locomotor and seven object manipulation skills (i.e., ball skill) on the TGMD-3. Previous studies assessed FMS performance in children with [48,54,55] and without disabilities [56,57,58], using the TGMD-3 in experimental intervention program settings to measure FMS focusing on locomotor and ball skills. The TGMD-3 was an important instrument for assessing the levels and progress of FMS in children with developmental disabilities (DD) [54,59,60]. Moreover, several studies used the TGMD for children with [54,60] and without disabilities [61,62] aged over 10 years old due to their lack of motor development, financial support, and facility use. The present study assessed the hop, horizontal jump, overhand throw, and dribble skills on the TGMD-3 to investigate locomotor and ball skills among children with DD through the exergame program. The TGMD-3 has thirteen FMS that consists of six locomotor and seven ball skills. The primary researcher selected four FMS (i.e., hop, horizontal jump, overhand throw, dribble) on the TGMD-3 that were two locomotor and two ball skills. Those FMS were crucial components of the tasks on the ICT-based exergame in the present study, also it was used to examine the impact of the ICT-based exergame on FMS performance among children with DD. The skill scores of each skill on the TGMD-3 were eight (hop), eight (horizontal jump), eight (overhand throw), and six (dribble), respectively. The performance videos of 52 children with DD were evaluated by two experts. Eligibility criteria for these experts were: (a) a Ph.D. degree in motor development or adapted physical education/activity, (b) assessing and rating the TGMD-3 among children with and without disabilities, and (c) a minimum of 5 years of experience teaching FMS content to children with DD in physical activity services. Expert raters independently scored the video. A total of three meetings were held to establish 100% agreement on the correct scores of four skills on the TGMD-3 in children with DD.

### 2.4. Procedure

The primary researcher designed a feasible exergame program considering the characteristics of children with DD. Based on previous studies that evaluated the motor abilities of children with DD, the following items were derived through discussions with the commissioned research director of the motor development and control major, a pediatric psychiatrist, and a professor specializing in physical education. The applicability of the ICT-based exergame program items was verified for children. Then, we evaluated items to examine the impact of the ICT-based exergame intervention on physical fitness and FMS performance among children DD for 12 weeks during a total of 18-month development period in Figure 4.

### 2.5. Data Analysis

The mean and standard deviation of the physical fitness and fundamental motor skill scores were calculated for the standing long jump, and two locomotor skills (i.e., horizontal jump, hop), and two ball skill scores (i.e., overhand throw, dribble) on the TGMD-3, respectively. The changes between the two groups (i.e., experiment and control groups) from the pre- to post-test were also analyzed using a two-way repeated-measures analysis of variance (Two-way RM-ANOVA) to compare the improvement of physical fitness and FMS in children with DD. All analyses were performed with IBM SPSS Statistics, version 28.0 (IBM Corp, Armonk, NY, USA). The significance level of the *p*-value was below 0.05.

## 3. Results

Table 2 presented mean scores and SDs of the participants’ body composition index between pre- and post-test in the present study. There were no significant differences in body composition indicators such as height, weight, and BMI between the groups while participating in the intervention program (*p* > 0.05; Table 3).

Muscular strength test (i.e., standing long jump): The comparison results of the muscular strength test and FMS between the experimental and control groups through pre- and post-test among children with DD are shown in Table 3. There was a significant improvement in the standing long jump test between the two groups (*F* (1, 50) = 18.79, *p* < 0.001). The distance records of the standing long jump changed from 88.3 ± 24.9 cm to 98.8 ± 29.3 cm. About 10 cm was increased on the standing long jump of children with DD using the Kinect and Intel Lidar sensor systems (see Figure 5). The hop skill score on the TGMD-3 significantly changed from 3.20 ± 1.70 to 5.08 ± 2.02 (*F* (1, 50) = 21.71, *p* < 0.001). Meanwhile, the horizontal jump skill score did not significantly change though it was slightly improved from 4.96 ± 2.11 to 5.76 ± 1.82 (*F* (1, 50) = 3.39, *p* = 0.07). Regarding the ball skill, both skills had significant *p*-values in the comparison between pre- and post-test of the groups on the TGMD-3 (overhand throw: *F* (1, 50) = 27.94, *p* < 0.001; dribble: *F* (1, 50) = 12.95, *p* < 0.001; see Table 3 and Figure 6).

## 4. Discussion

The purpose of this study was to develop an ICT-based exergame program for children with DD and to verify its effectiveness on their physical fitness and fundamental motor skills (FMS). In the developing phase of the exergame program, the primary researcher established the proper program content for children with DD and cooperated with the information communication and technology (ICT) research team, which involved various experts in computer science, psychology, biomechanics, adapted physical education, occupation therapy, and motor learning and control. The research team created a feasible ICT-based exergame program for children with DD. The findings in this study provided evidence to meet the suggestions in previous studies [7,8,43,63] that provided future directions for exergame activities for persons with disabilities.

The ICT-based exergame program which was developed for this study could be utilized for enhancing physical fitness and FMS among children with disabilities, especially DD. As shown in Figure 2 and Figure 3, the present ICT-based exergame program optimized the space setting widely with a front and ground screen for bringing up target objects and demonstrations by virtual characters. Furthermore, the movement of a participant or equipment, such as a bean bag and rubber ball, was tracked to show their performance either correctly or incorrectly. As mentioned above, the previous studies using existing exergames suggested flexible exergames with adjustable difficulty for easy play for individuals with disabilities [6,7,8]. The developed ICT-based exergame program could adjust difficulty levels and types of activity tasks according to the degree of cognitive and movement performance evaluation of each participant with DD. This ICT-based exergame program could be an initial technology model in not only developing exergame programs to promote physical fitness and FMS for children with DD but also encouraging their engagement in different types of exercise. Regarding physical fitness to test muscle strength in children with DD, there was a significant improvement in standing long jump through the ICT-based exergame program in the present study. It implied the ICT-based exergame program could be used as a functional alternative to actively engaging in PA like previous studies using a relevant exergame (i.e., Nintendo Wii) [64,65,66].

Instead of focusing on fitness as a single contributor to health, physical literacy places emphasis on the variety of components that are integral to being physically active throughout the life course [36]. For children with DD to become physically literate, they need sufficient opportunities to develop the competence to perform a repertoire of movement skills; however, repeated practice of individual skills in isolation is not sufficient. Children with DD also need to have the confidence to perform those skills in a variety of activities and contexts to engage in physical activities. For example, a child with DD must be able to strike, throw, catch, and run the bases to participate successfully in all aspects of a softball game with their peers. Learning to catch a ball from different distances and to a variety of ball speeds, sizes, and directions, will help to generalize “catching” to more aspects of a baseball game. Being able to perform a variety of FMS in a variety of contexts will afford a child more opportunities to participate with increased independence, to further develop their knowledge of a game or activity, and to understand the variety of ways in which their skills could be used through engagement in activities [36].

Fundamental motor skills (FMS) play a significant role in not only participating in games, sports, and leisure activities, especially for children [67] but also in daily physical activities [58]. There was a significant relationship between FMS proficiency and engagement in physical activity (PA) [68]. This meant the development of FMS competence showed a positive effect on active engagement in PA. During early childhood, participation in PA provides opportunities for children to develop FMS. They demonstrated that children can actively participate in a wider range of activities, which should increase participation and levels of PA when the majority of children are equipped with a repertoire of FMS. Competence of FMS in children could encourage their participation in PA. According to Bryanton and colleagues [69], a virtual reality (VR) game including physical activity showed significantly positive results in movement proficiency among children with cerebral palsy. This implied that VR or exergame programs might give a significantly positive impact on the development both of physical activity and FMS. It also would be helpful to promote physical fitness among persons with disabilities.

Many children with DD have greater difficulty with FMS performance than typically developing children [16,17,70]. It may result in hardship to progress FMS toward skillful performance after childhood. As such, the relationships among constructs should be considered age-related and based more so on the levels of motor competence demonstrated by children with DD. Movement is essential to participation, which in turn benefits health and levels of physical fitness [71,72]. There was a significantly improved level of physical fitness, such as muscle strength, and FMS (i.e., hop, overhand throw, dribble) in this study considering the reciprocal relationship between FMS and physical fitness. However, this study found that the standing long jump result to assess muscle strength had significant improvement between pre- and post-test among the participants throughout the ICT-based exergame intervention. Whereas, the horizontal jump skill on the TGMD-3 among those populations had no significant change. Although there is a reciprocal association between FMS and physical fitness in prior studies [71], a present study could not find the same result regarding the relationship between standing long jump ability and horizontal jump performance. This implies that some associations between FMS and physical fitness components may not show a positive relationship in children with DD, such as throwing distance and overhand throw skill performance, or standing long jump and horizontal jump skill performance. These assessments could be influenced by individual physical strength or detailed performance skills. For instance, the kicking point of a ball and a part of the foot could give an effect on the kick distance rather than the desired kick skill performance on the movement instrument (i.e., TGMD-3). Especially, the way to improve the horizontal jump skill performance should be to provide specific task instructions on the game demonstration for children with DD in alignment with the desired skill performance on the TGMD-3. Therefore, future research should be implemented to develop well-organized exergame content for enhancing both FMS and physical fitness for children with DD to promote their health. In addition, there was a suggestion of aerobic capacity influences that were about the intensity and duration of PA participation from FMS interventions [71]. According to discussed suggestions above, future research should be conducted to assess the cardiovascular ability of children with DD through ICT-based exergame programs.

The limitations of the present study are that it only assessed standing long jump test in muscular strength and four FMS among the thirteen FMS on the TGMD-3. Due to the diverse characteristics of children with DD and usage limits in the experimental setting, this study applied those variables to assess FMS and physical fitness among those participants. Therefore, the present study’s results may be different from previous studies using other exergames or video games. Related to the types of disability, this study only involved children with ASD and ID. Other types of DD among children may have different behavior and movement characteristics according to types of disabilities, such as down syndrome and cerebral palsy. Therefore, there is a limitation to generalizing the result of this study to all children with DD. Lastly, the ICT-based exergame program consisted of a limited number of physical activities, such as jumping, chasing, imitating, throwing, and kicking. Those activity contents had a limitation in offering a variety of movement performance activities, such as sliding, striking, and catching. Future ICT-based exergame programs should be developed to include those skill activities to improve FMS and physical fitness among children with DD.

## 5. Conclusions

This study had the purpose of developing an ICT-based exergame program for children with developmental disabilities (DD) and verifying its effectiveness on their physical fitness and fundamental motor skills (FMS). There were significantly positive impacts of the ICT-based exergame intervention on muscular strength and FMS in children with DD. The positive results included the following: (a) standing long jump, and (b) three FMS (i.e., hop, overhand throw, dribble) on the movement assessment. This ICT-based exergame program could improve muscular strength and targeted FMS to promote engagement in a variety of physical activities. However, future research for children with DD should be implemented not only to measure other components of physical fitness and FMS but also to cover the discussed limitations.

## Figures and Tables

**Figure 1 jcm-11-05890-f001:**
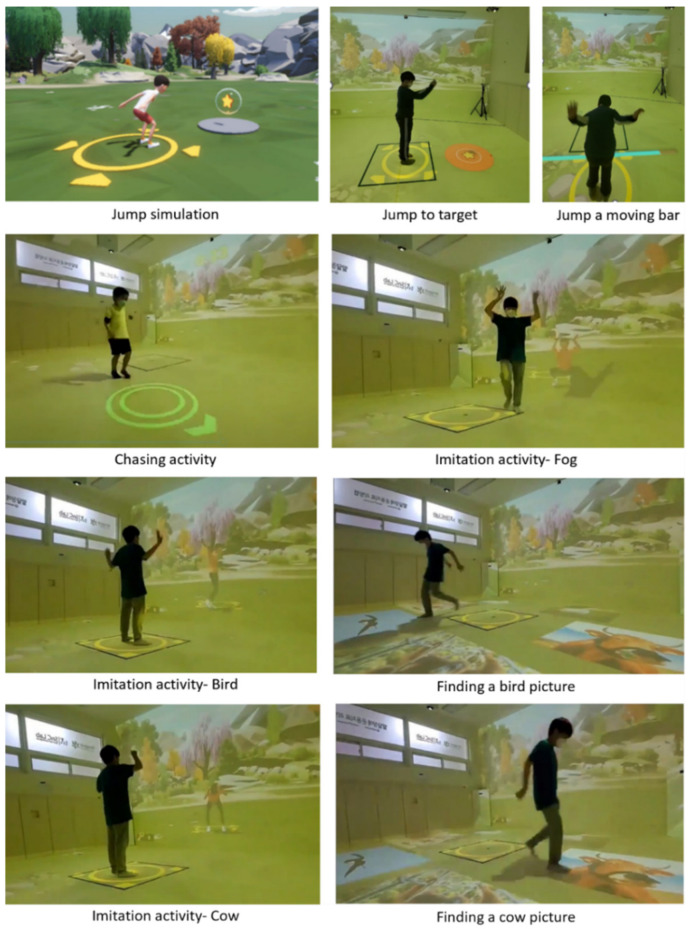
Locomotor skill simulation and actual execution in information and communications technology (ICT)-based exergame.

**Figure 2 jcm-11-05890-f002:**
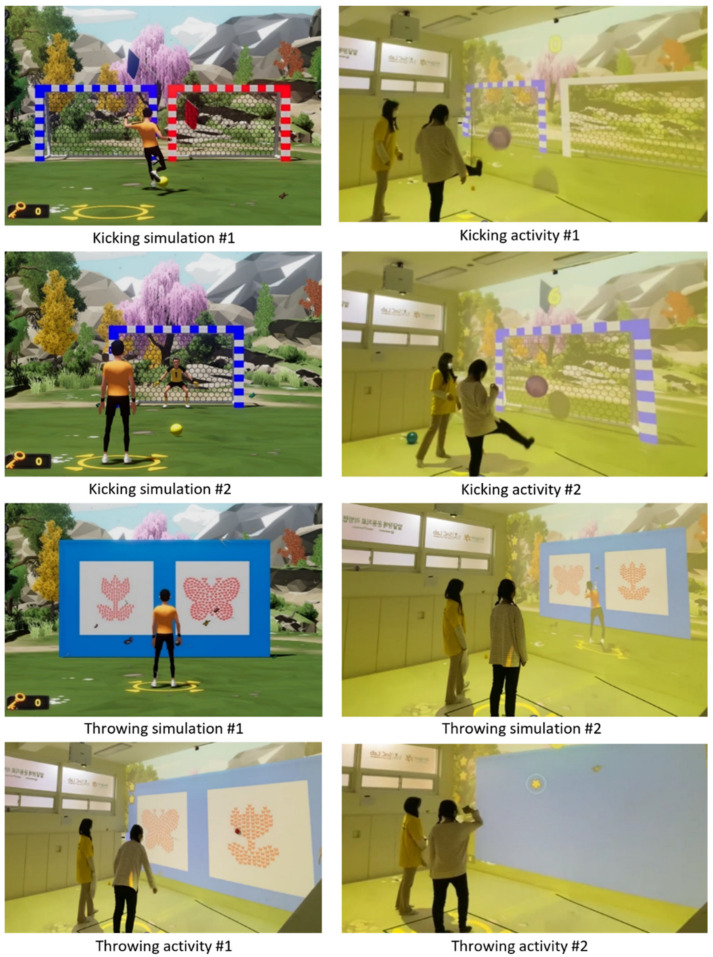
Ball skill simulation and actual execution in ICT-based exergame.

**Figure 3 jcm-11-05890-f003:**
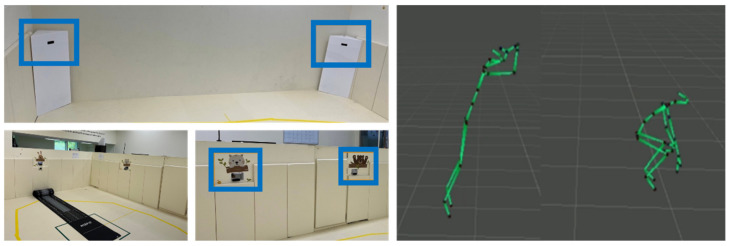
Azure Kinect and Intel Lidar setting for the standing long jump test.

**Figure 4 jcm-11-05890-f004:**
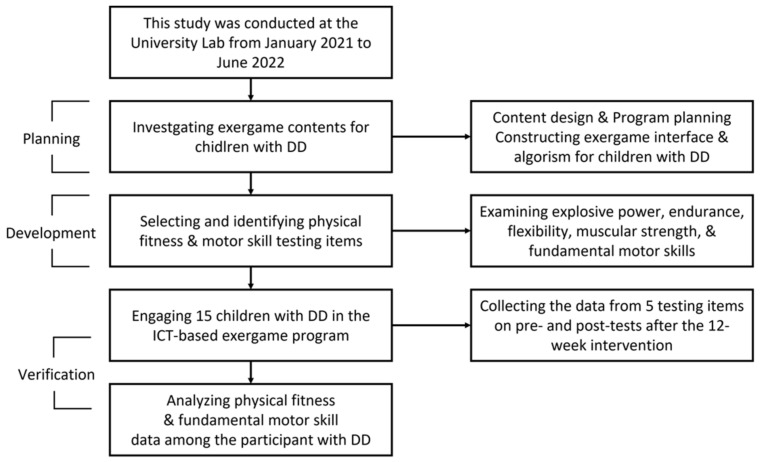
Flow chart of the study.

**Figure 5 jcm-11-05890-f005:**
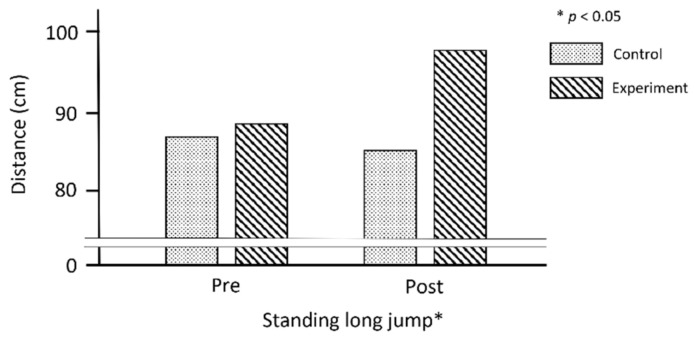
Muscle strength test results in physical fitness.

**Figure 6 jcm-11-05890-f006:**
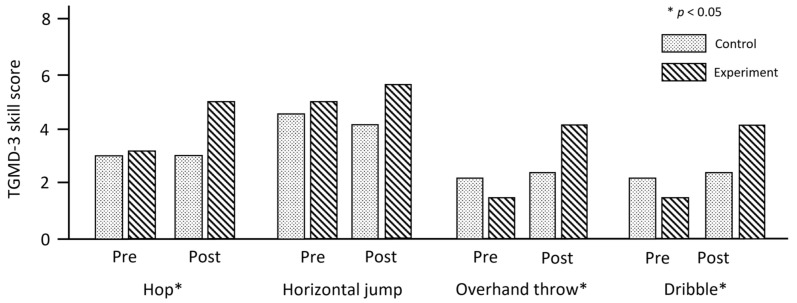
Fundamental motor skill test results.

**Table 1 jcm-11-05890-t001:** Functional evaluation lists in children with Developmental disabilities (DD).

Domain	Cognitive	Attractive	Psychomotor
Evaluation Subdomain	▪ Working memory ▪ Visual perception ▪ Executive function: plan, reaction, regulation, attention	▪ Following instructions ▪ Imitation ▪ Sharing ▪ Perception of emotion ▪ View acceptance	▪ Muscular strength ▪ Endurance ▪ Coordination ▪ Motor skill

**Table 2 jcm-11-05890-t002:** Demographic characteristics of the participants.

Category		Experiment Group			Control Group		
		ASD (*n* = 15)	ID (*n* = 12)	Total (*n* = 27)	ASD (*n* = 15)	ID (*n* = 10)	Total (*n* = 25)
Gender	Female	2	6	8	4	3	7
	Male	13	6	19	11	7	18
Age ^a^		10.21 ± 1.64	10.08 ± 1.32	10.15 ± 1.51	10.07 ± 1.39	10.51 ± 1.57	10.24 ± 1.48
Height (cm) ^a^	Pre-test	145.4 ± 9.5	141.9 ± 9.3	143.8 ± 9.6	139.9 ± 8.9	136.5 ± 4.9	138.6 ± 7.8
	Post-test	147.3 ± 9.3	144.6 ± 9.9	146.1 ± 9.7	141.9 ± 9.6	136.5 ± 4.9	138.6 ± 7.8
Weight (kg) ^a^	Pre-test	42.5 ± 7.6	38.9 ± 6.5	40.9 ± 7.4	33.6 ± 5.4	32.6 ± 3.9	33.2 ± 4.9
	Post-test	42.41 ± 8.1	38.9 ± 7.2	40.9 ± 7.9	34.9 ± 5.3	34.9 ± 3.6	34.9 ± 4.7
BMI ^a^	Pre-test	29.1 ± 4.3	27.3 ± 3.5	28.3 ± 4.1	33.6 ± 5.4	23.8 ± 2.2	33.2 ± 4.9
	Post-test	28.7 ± 4.7	26.7 ± 3.7	27.8 ± 4.4	24.6 ± 2.7	25.1 ± 1.9	24.8 ± 2.5

ASD (autism spectrum disorder); ID (intellectual disability); (a) Mean ± SD.

**Table 3 jcm-11-05890-t003:** Descriptive and statistical results in physical fitness and fundamental motor skills between pre- and post-test of the groups.

Category		Control		Experiment		*F* (1, 50)	*p*
		Pre-Test ^a^	Post-Test ^a^	Pre-Test ^a^	Post-Test ^a^		
Standing long jump (cm)		87.4 ± 23.1	85.6 ± 20.2	88.3 ± 24.9	98.8 ± 29.3	18.79	0.000 *
Locomotor skill	Hop	3.16 ± 1.19	3.16 ± 1.25	3.20 ± 1.70	5.08 ± 2.02	21.71	0.000 *
	Horizontal jump	4.68 ± 1.95	4.36 ± 1.49	4.96 ± 2.11	5.76 ± 1.82	3.39	0.072
Ball skill	Overhand throw	2.20 ± 1.44	2.52 ± 1.17	1.84 ± 2.11	4.44 ± 1.60	27.94	0.000 *
	Dribble	2.20 ± 1.39	2.44 ± 1.24	1.92 ± 1.55	3.92 ± 1.83	12.95	0.000 *

(a) Mean ± SD; * *p* < 0.001.

## Data Availability

The data presented in this study are available on request from the corresponding author. The data are not publicly available due to privacy.

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
