# Peer review of "Development of an ICT-Based Exergame Program for Children with Developmental Disabilities"

_jcm, 2022, doi:10.3390/jcm11195890_

Round 1

Reviewer 1 Report

Paper Title: Development of an ICT-based Exergame Program for Children with Developmental Disabilities.

This is an interesting study on exploring the effects of ICT-based exergame on Children with DD. Several problems are required to revise.

1.     Line 37, the word of “in- creasing” shall be delete the space.

2.     Line85, lack of social skills may be correct. Please change it.

3.     Authors used the precise exercise program in the experiment, why did not the children have an improvement in Horizontal jump?  But there is a reciprocal association between FMS and physical fitness in prior studies. 1) You mean that some studies showed the positive relationship? Or one study? 2) How many participants were included in the previous study? The sample size may influence the result. Additionally, previous study also used the standing long jump training, which was same as your current study. Therefore, authors should make a clear clarification. 

Author Response

Reviewer #1

Recommendation: Minor Revision

Reviewer #1_Comment 1:

Line 37, the word of “in- creasing” shall be delete the space.

Author’s response to Reviewer #1_Comment 1:

Appreciate your note. Deleted the space following Reviewer #1

Reviewer #1_Comment 2:

Line85, lack of social skills may be correct. Please change it.

Author’s response to Reviewer #1_Comment 2:

Thank you for the note. The typo was correctly changed to ‘lack of social skills’.

Reviewer #1_Comment 3:

Authors used the precise exercise program in the experiment, why did not the children have an improvement in Horizontal jump? 

Author’s response to Reviewer #1_Comment 3:

Thanks for your comment. We revised the sentence on the discussion part to present the improvement difficulty of the horizontal jump skill performance on the TGMD-3 among children with DD in Line 403 - 409.

Reviewer #1_Comment 4:

But there is a reciprocal association between FMS and physical fitness in prior studies. 1) You mean that some studies showed the positive relationship? Or one study?

Author’s response to Reviewer #1_Comment 4:

Thank you for the note. We revised the sentence to provide references about the association in Lile 155 – 159.

Reviewer #1_Comment 5:

2) How many participants were included in the previous study? The sample size may influence the result. Additionally, previous study also used the standing long jump training, which was same as your current study. Therefore, authors should make a clear clarification.

Author’s response to Reviewer #1_Comment 5:

Thank you for the note. There were 52 participants divided into 27 and 25 children with DD in the experiment and control group, respectively. Also, this manuscript showed the G*Power to support statistic results from the recruited number of samples in the participant section.

Associate Editor_Comment 6:

Additionally, previous study also used the standing long jump training, which was same as your current study. Therefore, authors should make a clear clarification.

Author’s response to Reviewer #1_Comment 6:

Thank you for the note. We revised this section to support using a standing long jump test in the present study with references.

We highlighted with comment responses in the revised manuscript to help your review. Thank you very much!

Reviewer 2 Report

Dear authors. The design of the study is very novel and interesting. I jus have some punctual comments in order to improve your conclusions. 1. The category of DD is too broad, as you say it correctly. Why did you chose this category? It would be more proper to include the children with specific motor difficulties and not the whole group of DD. You have to include some kind of critics in relation with such broad umbrella category. 2. Why did you chose virtual activities? As psychologist and neuropsychologist of cultural historical approach, I would say to you: is is better to play in the park and use real toys and real objects. Why did you prefer only virtual reality? All children need social communication with the pairs and with the adults, all children need orientation and real participation of the others in their developmental activities. 3. The group of the children is too broad, so that statistical differences make no real sense. It's importante to study concrete types of motor difficulties and arrange the groups not according to son broad categories in order to obtain sensible results.

Even so, I find the study interesting. I think you should be more critical, while presenting results, discussion and n conclusions. I suggest revise the literature of the authors from cultural historical psychology and neuropsychology.

Author Response

Reviewer #2

Recommendation: Minor Revision

Comments:

The category of DD is too broad, as you say it correctly. Why did you chose this category? It would be more proper to include the children with specific motor difficulties and not the whole group of DD. You have to include some kind of critics in relation with such broad umbrella category. 2.

Author’s response to Reviewer #2_Comment 1:

I appreciate your comment. We revised the sentence with the reason that the present study selected ID and ASD among DD to meet your comment.

Comment 2:

Why did you chose virtual activities? As psychologist and neuropsychologist of cultural historical approach, I would say to you: is is better to play in the park and use real toys and real objects.

Author’s response to Reviewer #2_Comment 2:

Thank you for the note. We added to describe the effect of VR, which was why this study chose VR as an intervention program for children with DD.

Comment 3:

Why did you prefer only virtual reality? All children need social communication with the pairs and with the adults, all children need orientation and real participation of the others in their developmental activities.

Author’s response to Reviewer #2_Comment 3:

Thanks for your comment. This study invented an exergame that included the components of VR and AR for children, especially DD. Children with DD have some limitations on psychological, cognitive, and movement development due to their characteristics. Many studies showed that VR and AR (i.e., exergame) gave positive effects on various areas of development among children with DD. Also, we believe those interventions may help to enhance their capabilities.

____________________________________________________________________________

Comment 4:

The group of children is too broad, so that statistical differences make no real sense. It's importante to study concrete types of motor difficulties and arrange the groups not according to son broad categories in order to obtain sensible results.

Even so, I find the study interesting. I think you should be more critical, while presenting results, discussion and n conclusions. I suggest revise the literature of the authors from cultural historical psychology and neuropsychology.

Author’s response to Reviewer #2_Comment 4:

I appreciate your comment. We would like to help your understanding of the group in this study. There were two groups divided into the experimental (n=27) and control groups (n=25). To help the understanding of the result of the study, we provided comparison figures 6 and 7 on how their results had been changed through the intervention.

Regarding the discussion part, we revised some sentences to provide critical insights to evaluate the result and future perspectives from the present study.

We highlighted with comment responses in the revised manuscript to help your review. Thank you very much!

Reviewer 3 Report

The article “Development of an ICT-based Exergame Program for Children with Developmental Disabilities” is well written and provides new tools for the development of physical fitness and motor competence in children with developmental disabilities.

Introduction

Minor comment: Line 37 change in-creasing into increasing

Materials and methods

Line 122-124: I don’t understand this phrase: “Specifically, game mechanics should be secure to understand the required performance in the development levels of children with DD.”

Different aspects of your study are combined into 1 sentence

Game mechanics / required performance / development of children with DD. More information is needed to combine these topics in one sentence and what do you want to explain here?

Line 162-165: Throwing a bean bag twice towards is a fundamental movement skill test  instead of a physical fitness test.

The border between physical fitness and motor competence is narrow, but in this study you present in table 3 correctly the different tests in the different constructs. 1 physical fitness test and 4 motor competence tests. In this section you should explain that you did not apply all tests of TGMD3 (only 4 tests were mentioned in table 3) and 1 test for physical fitness (standing long jump).

Line 264-275 is a repetition of line 217-228

Line 264-265: “Standing long jump is one of the measurement methods to evaluate reflexes” This is not correct, according to different authors and the council of Europe (1988) the standing long jump measures explosive strength (physical fitness), please modify this sentence.

Discussion

In my opinion, the authors generalize the results, but only measured a limited number of variables. Please adapt the discussion in this way.

Line 355-356: I don’t understand the sentence “That is a fact the powerful link between engagement in physical activity and FMS proficiency”

Something is missing in this reasoning

Line 368-369: “and progress towards mastery of these skills after childhood” is this an assumption? You did not you have not demonstrated this in this research.

Please adapt thes minor comments and i wish to congratulate the authors with their nice research

Author Response

Reviewer #3

Recommendation: Minor Revision

Comments 1: Minor comment: Line 37 change in-creasing into increasing

Author’s response to Reviewer #3_Comment 1:

Thanks for your note. We revised the space on the sentence.

Comment 2:

Line 122-124: I don’t understand this phrase: “Specifically, game mechanics should be secure to understand the required performance in the development levels of children with DD.”Different aspects of your study are combined into 1 sentence. Game mechanics / required performance / development of children with DD. More information is needed to combine these topics in one sentence and what do you want to explain here?

Author’s response to Reviewer #3_Comment 2:

I appreciate your note. We revised the sentence to help understand about the considerations of the intervention for children with DD.

Comment 3:

Line 162-165: Throwing a bean bag twice towards is a fundamental movement skill test  instead of a physical fitness test.

The border between physical fitness and motor competence is narrow, but in this study you present in table 3 correctly the different tests in the different constructs. 1 physical fitness test and 4 motor competence tests. In this section you should explain that you did not apply all tests of TGMD3 (only 4 tests were mentioned in table 3) and 1 test for physical fitness (standing long jump).

Author’s response to Reviewer #3_Comment 3:

Thanks for your note. We revised the place of throwing a bean bag to the fundamental motor skill section. Additionally, we provided another task on the physical fitness part.

Comment 4:

Line 264-275 is a repetition of line 217-228

Author’s response to Reviewer #3_Comment 4:

Really appreciate your note. We deleted the redundant section.

Comment 5:

Line 264-265: “Standing long jump is one of the measurement methods to evaluate reflexes” This is not correct, according to different authors and the council of Europe (1988) the standing long jump measures explosive strength (physical fitness), please modify this sentence.

Author’s response to Reviewer #3_Comment 5:

Thanks for your comment. We revised the sentence to meet your comment in Line 261-266.

Comment 6:

Line 355-356: I don’t understand the sentence “That is a fact the powerful link between engagement in physical activity and FMS proficiency”

Something is missing in this reasoning

Author’s response to Reviewer #3_Comment 6:

Thank you for the note. We revised the sentence to describe the relationship between PA and FMS competence in Line 366-368.

Comment 7:

Line 368-369: “and progress towards mastery of these skills after childhood” is this an assumption? You did not you have not demonstrated this in this research.

Author’s response to Reviewer #3_Comment 7:

We appreciate your comment. We revised the sentence to provide better understanding of the movement characteristics among children with DD in Line 379-380.

We highlighted with comment responses in the revised manuscript to help your review. Thank you very much!
